# The Synthesis of SNAC Phenolate Salts and the Effect on Oral Bioavailability of Semaglutide

**DOI:** 10.3390/molecules29163909

**Published:** 2024-08-19

**Authors:** Tovi Shapira-Furman, Ayala Bar-Hai, Amnon Hoffman, Abraham J. Domb

**Affiliations:** Institute for Drug Research, School of Pharmacy, Faculty of Medicine, The Hebrew University of Jerusalem, P.O. Box 12065, Jerusalem 91120, Israel; tovi.furman@mail.huji.ac.il (T.S.-F.); ayala.fraenkel@mail.huji.ac.il (A.B.-H.); amnonh@ekmd.huji.ac.il (A.H.)

**Keywords:** phenol, salt synthesis, SNAC, semaglutide

## Abstract

Purpose: Sodium N-[8-(2-hydroxybenzoyl)amino]caprylate (SNAC) is a well-known penetration enhancer widely used in commercial applications. This study aims to broaden its properties through a novel strategy of converting it into its phenolate salts. The objective is to investigate the synthesis of SNAC phenolate salts, specifically SNAC–choline (SNAC-CH), SNAC–sodium (SNAC-Na), and SNAC–phosphatidylcholine (SNAC-PC), and to explore their potential application in improving the oral absorption of semaglutide. Methods: The synthesis of SNAC phenolate salts was confirmed through 1H-NMR, FTIR, and an elemental analysis of C, H, N, and O. In vivo testing was conducted to assess the oral delivery of semaglutide using these synthesized SNAC phenolate salts. Pharmacokinetic (PK) values were measured to evaluate the impact on drug absorption. Results: The synthesis of SNAC phenolate salts (SNAC-CH, SNAC-Na, and SNAC-PC) was successfully achieved under appropriate conditions, and their structures were confirmed using analytical techniques such as IR, NMR, and CHN elemental analysis. The paradigm of their use was evaluated through an oral pharmacokinetic (PK) in vivo study using SNAC phenolate salts, which did not impair the original SNAC PK values. This suggests that this strategy holds promise as a potential new effective enhancer for oral absorption. Conclusions: The utilization of SNAC phenolate salts presents a novel and promising strategy for extending the verity of penetration enhancers’ molecules and properties. Synthesizing phenolate salts represents a new chemical strategy that may open new avenues in molecular development. This approach holds future potential to enhance the oral delivery of peptide drugs like semaglutide without compromising therapeutic efficacy. Overall, it offers significant advancements in the field by providing a potential alternative to injectable peptides through oral delivery systems.

## 1. Introduction

Sodium N-[8-(2-hydroxybenzoyl) amino] octanoate, known as SNAC or salcaprozate sodium, is a synthetic N-acetylated amino-acid derivative of salicylic acid. It contains an aromatic structure—phenol, a fatty chain and carboxylate sodium salt. It was first synthesized during a molecular screening aimed at optimizing penetration enhancers in order to formulate them with bioactive compounds to enhance their bioavailability [1]. Chemical variation is commonly achieved by modifying factors such as the length of the fatty chain, the type of carboxylate salt, etc. These modifications introduce new properties depending on the chemical groups present within the molecule. The SNAC molecule contains a phenolic group, as mentioned previously.

The medicinal utility of phenols extends across various therapeutic areas, including but not limited to analgesics, anti-inflammatories, antimicrobials, antivirals, and anticancer agents [2]. While converting carboxylic acids into salts is widely used in medicinal chemistry—as most amine or carboxylic acid-based pharmaceutical compounds are available in salt form [3,4]—only a few indications exist for phenolate salt formation in this field [5]. The phenolic group is a weak acid moiety that has the ability to be converted into its salt form through an acid–base reaction. This modification can import advanced attributes into the substance, as regards to solubility in water and lipids, hydrophilicity, and physical properties [6]. Phenolate salts form ion pairs with a counter-ion of the opposite charge. The nature of the counter-ion contributes new properties to the converted molecule.

Amongst researchers, peptide drugs have attracted significant interest, since peptides portend a promising future for therapeutic interventions by closely mimicking natural biological pathways [7]. More than 60 peptide drugs have been granted clinical approval by the US FDA; nonetheless, in light of their high enzymatic sensitivity and limited permeability, a majority of these drugs are administered via parenteral routes. Multiple injections are costly and inconvenient, particularly in chronic treatment, potentially leading to reduced adherence and poor patient compliance [8]. Numerous strategies have been investigated to resolve these challenges, such as encapsulation into nanoparticles, liposomes, and microspheres [1]. Many of these approaches, however, are complicated by inherent problems, including low yield and loading, costly production, and issues in process scale up. Permeation enhancers containing medium-chain fatty acids are often part of the development of protein and peptide delivery systems. This is because of their ability to improve intestinal permeability and their relative facility to be incorporated into formulations. SNAC is an example of the most clinically tested permeation enhancers for the oral delivery of bioactive agents. It was first used for the oral delivery of vitamin B_12_ (Eligen©) and was assigned the status of being generally recognized as safe (GRAS) [9]. A pharmacokinetics clinical trial comparing oral vitamin B_12_ and oral formulation with SNAC revealed a significantly improved bioavailability for the SNAC formulation, as well as reduced T_max_ [9]. The mechanism of action associated with the permeation of the substance is not fully understood, but evidence suggests that tight junctions are not involved in the mechanism of action of SNAC. Consequently, SNAC is classified as a transcellular permeation enhancer [10]. Researchers have suggested that better penetration may be achieved through the amplification of hydrophobicity via non-covalent interactions between SNAC and the macromolecule. This interaction exposes hydrophobic regions, which facilitate transcellular permeation. A different hypothesis puts forth complexation between the conjugated base on SNAC that forms in the pH environment of the small intestine and the basic amino acids of the peptide that insert together through the membrane [11]. Further absorption pathways have been suggested by the study of numerous proteins and peptides formulated with SNAC [12]. Recently, the FDA approved an oral semaglutide product, a tablet with SNAC as the absorption enhancer: RYBELSUS^Ⓡ^ (Novo Nordisk, Copenhagen, Denmark). Semaglutide is a human Glucagon-like peptide-1 receptor agonist (GLP-1RA) known for its use in the treatment of type 2 diabetes, as it effectively controls glycemic levels, thus promoting weight loss and improving cardiovascular results in patients [8]. This product constitutes a significant milestone in the development of the first orally administered GLP-1RA, providing patients with an alternative to conventional injection-based treatment. Buckley et al. conducted a comprehensive study of the absorption mechanism of semaglutide administered via SNAC tablet. They arrived at the unforeseen conclusion that absorption primarily occurs in the stomach. This finding is unexpected, since the primary site for absorption of the gastrointestinal tract is typically the small intestine, due to its extensive absorptive surface area provided by the microvillus structure of the epithelial mucosae [13]. Their research also unveiled a distance to the semaglutide concentration relationship, with the highest concentration observed in close proximity to the tablet. This suggests that the tablet settles in the stomach and undergoes erosion, enabling semaglutide to be released in the closest proximity possible to the absorption site and minimizing enzymatic degradation. The SNAC responsible for pH elevation also surrounds the delivered compound, yielding activity suppression of pepsin, the most common protease enzyme [1,13]. Even so, it should be noted that a significant amount of SNAC is needed for this purpose. Despite these efforts, the reported absorption of oral semaglutide remains low (<1%) [14]. Thus, there is a continuing need for additional and relevant formulation improvements.

The aim of this research was to synthetize novel phenolate salts. As a paradigm, we investigated the effect of SNAC phenolate salts on the bioavailability of semaglutide and sought to develop new potential approaches for peptide oral formulations.

In this investigation, we took advantage of the presence of reactive phenol to synthesize various SNAC salts, which are anticipated to possess unique characteristics distinct from the original molecule. The approach aimed to convert the known penetration enhancer SNAC into novel phenolate salts. As a proof of concept, we also examined whether it affects the bioavailability of semaglutide following oral administration in vivo. Our findings demonstrate the formation of three different SNAC salts. Pharmacokinetic studies using these salts showed similar or even improved pharmacokinetic values compared to the original SNAC, and a reduced required amount for SNAC, therefore opening up new possibilities for the further development of injectable proteins and peptides and their transformation into tablet-based oral medicine.

## 2. Methods

General methods:

FTIR was performed on a Nicolet iS10 spectrometer (Thermo Fisher Scientific, Waltham, MA, USA) using a Smart iTR sampling accessory. ^13^C- and ^1^H-NMR spectra were obtained on a Varian 300 MHz (Palo Alto, CA, USA) with samples dissolved in DMSO-d_6_. The determination of C, H, N, and O was performed using the Thermo Flash 2000 CHN-O Elemental Analyzer (Thermo Fisher Scientific, Waltham, MA, USA).

### 2.1. Salts Synthesis

SNAC is a carboxylic acid sodium salt. In this study, the term “salt” refers to the phenolic proton substitution to a new form: sodium, choline, or phosphatidylcholine.

#### 2.1.1. SNAC-Na Salt

SNAC (100 mg, 0.33 mmol, 1 eq) was dissolved in 6 mL methanol containing NaOH (14.6 mg, 0.33 mmol, 1 eq) and mixed overnight at room temperature, under nitrogen atmosphere. Methanol was removed, and the formed SNAC-Na salt was purified with n-heptane, to yield 108 mg (95%). FTIR: νmax/cm^−1^ 2928–2858 cm^−1^ (C–H stretching), 1618 cm^−1^ (amide carbonyl stretching), 1568 cm^−1^ (carboxylate carbonyl stretching),1452 cm^−1^ (C–N stretching), 1424–1406 cm^−1^, (C=C stretching), 1337–1300–1261–1215–1145 cm^−1^ (C–O stretching), 1030 cm^−1^ (C–C bending), and 881–885–819–763–726–703 cm^−1^ (C–H bending). ^1^H NMR (300 MHz, DMSO-d_6_) δ 12.37 (s, 1H), 7.54 (dd, *J* = 7.7, 2.1 Hz, 1H), 6.80 (ddd, *J* = 8.6, 6.7, 2.1 Hz, 1H), 6.22 (d, *J* = 8.3 Hz, 1H), 5.97 (t, *J* = 7.2 Hz, 1H), 3.18 (q, *J* = 6.2 Hz, 2H), 1.87 (t, *J* = 7.3 Hz, 2H), 1.41 (p, *J* = 6.5, 6.1 Hz, 4H), and 1.25 (d, *J* = 4.3 Hz, 6H). ^13^C NMR (75 MHz, DMSO-d_6_): δ 178.02, 172.01, 168.89, 131.39, 129.73, 121.92, 119.11, 108.85, 38.64, 38.32, 29.72, 29.18, 27.10, 26.75, and 25.89.

#### 2.1.2. SNAC–Choline Salt

Following the general procedure of SNAC-Na, the SNAC–choline salts, either 1:1 or 1:2, were achieved as clear yellow color solid by conducting the reaction between SNAC (100 mg, 0.33 mmol, 1 eq) and choline hydroxide (46 wt% solution in water, 81 µL, 0.33 mmol, 1 eq) or (46 wt% solution in water, 162 µL, 0.66 mmol, 2 equiv.). Yield: 136.4 mg (96%). FTIR: νmax/cm^−1^ 3246 (choline–OH stretching), 2926–2854 cm^−1^ (C–H stretching), 1613 cm^−1^ (amide carbonyl stretching), 1557 cm^−1^ (carboxylate carbonyl stretching), 1459 cm^−1^ (C–N stretching), 1403–1332 cm^−1^, (C=C stretching), 1250–1199–1145 cm^−1^ (C–O stretching), 1088–1055 cm^−1^ (C–C bending), and 952–863–762-703cm^−1^ (C–H bending). ^1^H NMR (300 MHz, DMSO-d_6_) δ 12.37 (t, *J* = 5.5 Hz, 2H), 7.57 (dd, *J* = 7.8, 2.1 Hz, 3H), 6.83 (ddd, *J* = 8.6, 6.7, 2.1 Hz, 3H), 6.29–6.20 (m, 3H), 6.06–5.95 (m, 3H), 5.50 (s, 1H), 3.83 (q, *J* = 3.1 Hz, 6H), 3.37 (dd, *J* = 6.3, 4.0 Hz, 6H), 3.20 (q, *J* = 6.2 Hz, 7H), 3.00 (s, 1H), 1.88 (t, *J* = 7.3 Hz, 6H), 1.43 (t, *J* = 7.4 Hz, 12H), 1.31 (d, *J* = 7.8 Hz, 7H), and 1.26 (s, 8H). ^13^C NMR (75 MHz, DMSO-d_6_) δ 177.65, 168.19, 167.95, 131.86, 129.86, 120.58, 119.08, 111.88, 67.56, 64.54, 60.40, 55.55, 54.57, 38.45, 38.25, 29.84, 29.55, 28.89, 26.85, and 26.47.

#### 2.1.3. SNAC–Phosphatidylcholine Salt

In a typical experiment, SNAC-Na (100 gm, 0.31 mmol, 1 eq) and phosphatidylcholine (235 mg, 0.31 mmol, 1 eq or 758.1 mg, 0.62 mmol, 2 eq) were mixed overnight in methanol under nitrogen. The SNAC–phosphatidylcholine was purified by washing with n-heptane and drying under high vacuum overnight. SNAC–phosphatidylcholine was obtained as a pale yellow-colored solid and yielded 346.3 mg (95%). FTIR: νmax/cm^−1^ 2982–2852 cm^−1^ (C–H stretching), 1734 (ester carbonyl stretching), 1639 cm^−1^ (amide carbonyl stretching), 1590 cm^−1^ (carboxylate carbonyl stretching), 1493 cm^−1^ (C–N stretching), 1446–1340 cm^−1^ (C=C stretching), 1242–1171cm^−1^ (C–O stretching), 1090–1060 cm^−1^ (strong bands PO_3_^2-^ and C–O bending), and 967–916–804–753 cm^−1^ (C–H bending). ^1^H NMR (300 MHz, DMSO-d_6_) δ 10.50 (s, 1H), 8.19 (s, 2H), 8.11 (dd, *J* = 6, 3 Hz, 2H), 7.35–7.29 (m, 2H), 7.03 (d, *J* = 9 Hz, 2H), 6.74 (t, *J* = 9 Hz, 2H), 5.53 (q, *J* = 6.4 Hz, 2H), 4.85 (s, 2H), 4.39 (t, *J* = 11.9, 6.2 Hz, 1H), 4.13–4.01 (m, 1H), 3.99–3.90 (m, 1H), 3.56 (s, 4H), 3.12–2.87 (m, 5H), 2.50 (td, *J* = 7.4, 4.5 Hz, 1H), 2.40–2.20 (m, 2H), 2.21 (d, *J* = 6.7 Hz, 1H), 1.87–1.65 (m, 1H), 1.75 (s, 2H), 1.60–1.41 (m, 13H), and 1.17–0.99 (m, 2H). ^13^C NMR (75 MHz, DMSO-d_6_) δ176.89, 173.00, 172.75, 167.76, 163.64, 132.53, 130.17, 129.60, 128.22, 119.02, 118.52, 115.39, 99.98, 71.02, 70.96, 65.96, 62.93, 58.79, 53.59, 38.28, 36.87, 34.05, 33.86, 31.77, 31.38, 29.56, 29.07, 28.88, 27.63, 25.81, 25.46, 24.87, 22.45, and 14.38.

### 2.2. Formulation Preparation

Semaglutide formulation was prepared using SNAC or SNAC-Na, SNAC-CH, and SNAC-PC. The formulations were prepared by gradual mixing of the following components: semaglutide (2.4% *w*/*w*), SNAC or SNAC salt (70.7% *w*/*w*), polyvinylpyrrolidone PVP K90 (1.9% *w*/*w*), Avicel ph101(23.6% *w*/*w*), and magnesium stearate (1.4% *w*/*w*). All compounds are powders.

IV solution was prepared by dissolving semaglutide in water containing disodium phosphate (0.014% *w*/*v*), propylene glycol (0.144% *w*/*v*), and phenol (0.056% *w*/*v*). The solution was diluted by 3 to reach semaglutide concentration of 0.046mg/mL and filtrated by a sterile 0.22 μm syringe filter.

### 2.3. In Vivo Study

Male Wistar rats (Harlan, Jerualem, Israel) were used in vivo. Each SNAC or salt formulation impacted the semaglutide following its administration.

Pharmacokinetic study protocol: Male Wistar rats (275–300 g, Harlan, Israel) were kept under a 12 h light/dark cycle, with free access to food and water. Animals were anesthetized, and an indwelling cannula was placed into the right jugular vein for systemic blood sampling. The cannula was tunneled beneath the skin and exteriorized at the dorsal part of the neck. The animals were transferred to individual cages to recover overnight (12–18 h). Throughout the experiments, free access to food was available 4 h post-oral administration.

Animals were randomly assigned to five groups: semaglutide with either SNAC, SNAC-Na, SNAC-CH, SNAC-PC, or no SNAC. Oral SNAC or salt formulations were dispersed in distilled water and then administrated by oral gavage (~1.2 mL for a rat to get semaglutide dose of 12 mg/kg). Blood samples (0.36 mL) were obtained by intravenous cannula, placed into the jugular vein. In the case of oral administration, samples were taken at 5 min pre-dose intervals and at different time points post-dose, according to the pharmacokinetic profile (no more than 10% of rat blood volume was drawn). To prevent dehydration, physiological solutions were administered, following each blood sampling. Plasma was separated by centrifugation (4000 rpm, 7 min, 4 °C) and stored at −20 °C until use.

Plasma samples were analyzed by a developed LC-MS method for semaglutide.

Absolute bioavailability values and other pharmacokinetic observations were assessed by comparison to intravenous bolus of semaglutide.

### 2.4. Sample Analysis

Semaglutide in plasma was performed in an Eppendorf^®^ tube by mixing 150 μL of the plasma sample with 150 μL acetonitrile (ACN) and 15 μL of 5% formic acid (FA) and vortexed for 5 min. The tube was centrifuged at 15 °C, 6000 rpm for 6 min. Then, 160 μL of the supernatant was collected in a new tube. A second extraction was performed for the remaining precipitant by adding 60 μL of 5% FA, followed by mixing and centrifugation. Supernatant, 50 μL, was transferred into a tube. The solvent was evaporated to dryness, reconstituted with 50 μL 5% FA, and analyzed by an LC-MS system. Semaglutide standards were prepared by spiking semaglutide solutions in 5% FA into plasma to obtain final concentrations of 1–0.005 μg/mL. The same extraction procedure was performed on the standards as well.

### 2.5. LC-MS Analytical Method

MS/MS analyses were determined on a Sciex (Framingham, MA, USA) QTRAP^®^ 6500^+^ mass spectrometer). The chromatography analysis was performed using reverse-phase conditions using a Shimadzu (Kyoto, Japan) UHPLC System, consisting of a Shimadzu CBM-20A communication bus module, Nexera X2 LC-30AD pump, including a Shimadzu DGU-20A5R degasser, a Shimadzu SIL-30AC autosampler, and a Shimadzu CTO-20AC column oven. Liquid chromatography separation was obtained using 10 μL injections of samples onto an XBridge Peptide BEH C18 column (3.5 µm 300Å, 50 × 2.1 mm) from Waters Corp. (Milford, MA, USA), protected by a SecurityGuard™ (Phenomenex, Torrance, CA, USA) cartridge (C18, 2 × 2.1 mm). The column was maintained at 40 °C during the entire analysis. Gradient elution mobile phases consisted of 0.1% formic acid in water (Phase A) and 0.1% formic acid in methanol (Phase B). A flow rate of 0.2 mL/min was applied with a linear gradient from 60% to 80% B over 3.5 min and then held at 80% B for additional 4.5 min. The column was then equilibrated for 6 min at starting conditions prior to the injection of the next sample. The gradient elution program (0.2 mL/min) is described in Table 1.

Semaglutide was detected in a positive ion mode using electron spray ionization (ESI) and a multiple reaction monitoring (MRM) mode of acquisition. The molecular ion of the semaglutide [M + 4H]^4+^ was selected in the first mass analyzer and fragmented in the collision cell, followed by the detection of the products of fragmentation in the second analyzer.

Optimal detection conditions were determined by constant infusion of a 200 ng/mL solution of semaglutide in methanol/water (8:2) using the integrated syringe pump (5 μL/min). Transitions were selected, and their settings were determined using Analyst 1.7.1 Software in compound optimization mode.

The IonDrive^TM^ Turbo V source temperature was set at 500 °C with the ion spray voltage at 5500 V. The curtain gas was set at 35.0 psi. The nebulizer gas (Gas 1) was set at 50 psi, and the turbo heater gas (Gas 2) was set at 60 psi. The entrance potential (EP) was set at 10 V. The dwell time was set at 30 ms. The collision energy (CE), de-clustering potential (DP), and collision cell exit potential (CXP) for the monitored transitions are given in Table 2.

Data acquisition was performed on a Dell Optiplex XE2 computer using Analyst 1.7.1, and the data were analyzed using Sciex OS Software, both distributed by Sciex.

### 2.6. Statistical Analysis

All values are expressed as mean ± standard deviation (SD). To determine statistical significance, one-way ANOVA, followed by Tukey’s test, was applied. A *p*-value less than 0.05 was termed significant.

## 3. Results and Discussion

The SNAC molecule contains a phenolic group that is highly chemically reactive, so it can be converted into a phenolate ion. SNAC salts were synthesized using NaOH, choline hydroxide, or phosphatidylcholine. Figure 1 represents the synthesis of SNAC salts from SNAC and sodium hydroxide, choline hydroxide, and phosphatidylcholine. SNAC is known as an absorption enhancer formulated with various substances to improve oral bioavailability [1,10]. SNAC is being used to address the oral bioavailability of protein and peptides, which, for now, are mainly administrated via a process involving multiple injections, constituting a less convenient method. Even though SNAC has been investigated in different oral protein delivery systems, only two forms have been used as clinical products. Improving the oral bioavailability of proteins requires a significant quantity of SNAC for a relatively small dose of the therapeutic protein with low detected levels of absorbed protein in the bloodstream [14]. The current study aims to develop an innovative class of SNACs, designed to emulate the performance of the conventional SNAC, readily available clinical-grade products, using the novel strategy of conversion phenol into salts. Transforming a substance into its salt form is a known technique in delivery system development, as it may improve its properties. Suitable salt forms may provide the required characterizations, such as increased lipophilicity, buffering effects, surfactants properties, and solubility, thereby presenting many therapeutic effects [4,6,13]. Recent studies have demonstrated the advantages of converting bioactive compounds into their phenolate salts, showing effective therapeutics outcomes [2,5]. Here, we present a similar strategy for formulating the agent.

### 3.1. SNAC Salts’ Synthesis and Characterization

SNAC salts were synthesized from sodium (Na) and choline through a dehydration reaction involving SNAC and the corresponding bases, NaOH and choline hydroxide. This reaction took place in the presence of methanol at room temperature under a nitrogen atmosphere, and the reaction proceeded overnight. Additionally, a SNAC salt, phosphatidylcholine, was prepared by means of a metathesis reaction between SNAC-Na and phosphatidylcholine under similar conditions of methanol, room temperature, and a nitrogen environment (Figure 1). All of these salts were produced with either 1:1 or 1:2 molar ratios of SNAC to the respective bases or phosphatidylcholine.

To comprehensively characterize the SNAC salts, both Nuclear Magnetic Resonance (NMR) spectroscopy and Fourier-Transform Infrared Spectroscopy (FTIR) were employed.

SNAC salts, namely Na^+^, choline hydroxide, and phosphatidylcholine salts, were investigated by NMR. The proton NMR spectra of the SNAC salts were compared to those of unmodified SNAC in DMSO-d_6_ at room temperature (as depicted in Figure 1). While the unmodified SNAC displayed aromatic hydroxyl resonances at 5.0 ppm, these resonances were absent in the SNAC phenolate salts, indicating phenolate salt formation. Moreover, several aromatic and aliphatic protons of the SNAC salts exhibited up-field shifts, compared to SNAC. In the case of SNAC–choline and SNAC–phosphatidylcholine salts, the presence of counter cations was confirmed through the observation of proton in NMR spectra at 3–5.5 ppm and 8.5–1 ppm, respectively.

SNAC salts were analyzed by FTIR. Characteristic bands corresponding to the aromatic hydroxyl group were identified at 3353 cm^−1^ (as shown in Figure 2 and Table 3). However, these peaks were no longer evident in the SNAC salts, with the exception of SNAC–choline, providing further evidence of SNAC’s transformation into an anionic form. In the case of SNAC–choline, the FTIR band at 3224 cm^−1^ was attributed to the presence of a hydroxyl functional group in the choline hydroxide. Furthermore, the stretching frequencies of alkyl C–H bonds in these compounds were identified within the range of 2852–2982 cm^−1^. The SNAC salts also exhibited stretching frequencies that are indicative of major functional groups within the following ranges: carbonyl in amide bond (1613–1639 cm^−1^), carbonyl in carboxylate bond (1557–1590 cm^−1^), C-N (1452–1493 cm^−1^), and C=C (1332–1446 cm^−1^), and C-O (1145–1261 cm^−1^). The vibrational frequencies displayed subtle shifts compared to the unmodified SNAC; however, they still reflected the presence of the respective functional groups in the SNAC salts.

The regenerative ability of SNAC from SNAC salts was investigated in acidic aqueous solutions with a pH range from 1 to 3. Using this method, a known quantity of SNAC salt was treated with 10 mL of acidic aqueous solutions with pH values ranging from 1 to 3. This resulted in the conversion of SNAC salts back into water-insoluble SNAC through protonation. The regenerated SNAC was then extracted from the aqueous layer using organic solvents, such as chloroform, and its chemical structure was confirmed using NMR spectroscopy. The studies concluded that SNAC salts have the capacity to revert to SNAC through protonation in an acidic medium.

The percentages of C, H, N, and O in SNAC salts were analyzed via an elemental analysis and compared with pure SNAC. The elemental composition of SNAC salts closely aligned with the theoretical values of the proposed structures (Table 4). According to the elemental analysis data, the Na atoms in the Na salt of SNAC were counterbalanced by a single SNAC anionic unit, accompanied by coordinated lattice water molecules. Likewise, the CHNO data for the SNAC salt of choline matched the structure consisting of one SNAC anionic unit per choline cation, along with one lattice water molecule. The data for the phosphatidylcholine salt of SNAC confirmed a structure comprising one SNAC anion per two phosphatidylcholine cationic units, without any lattice water molecules. While the metal salts of SNAC fit well with the theoretical values of the CHNO elemental analysis, the elemental analysis of the phospholipid salt indicates excess phospholipid, which is probably related to the large phospholipid residue that forms partial salts of the SNAC carboxylate and phenolate sites for salt formation and the difficulty to isolate the excess phospholipids from the salt product.

### 3.2. In Vivo PK Study

A SNAC molecule is known as an absorption enhancer, and it is used clinically [9]. Here, the oral bioavailability of semaglutide was selected after formulating it with SNAC or SNAC salts. Accomplishing oral bioavailability is a great challenge, particularly regarding protein and peptide delivery. This is due to their large size and low stability in the GI environment [15]. Figure 3 exhibits semaglutide blood concentrations during post-oral administration to freely moving rats. It is clear that the administration of semaglutide not including SNACs led to negligible semaglutide absorption, emphasizing the essential role of SNACs in reaching peptide bioavailability. The conversion of SNAC into salts did not compromise its activity, as shown by the absence of significant differences observed between the SNAC formulation profile and other SNAC formulations. The plasma concertation vs. time plot of SNACs is characterized by a prolonged absorption phase. This observation suggests a promising potential for a modified semaglutide formulation based on SNAC, characterized by a moderate terminal slope and a long-lasting effect. PK parameters were calculated from the semi-logarithmic data (Table 5), revealing that SNAC-CH and SNAC-Na reached similar values in terms of area under the curve (AUC), Cmax, and %F, while SNAC-PC presented an ~3 fold increase in AUC_0−8_, ~4 fold increase in AUC_0−∞_, and ~4 fold increase in F%, indicating a higher absolute bioavailability than the original SNAC formulation, significantly. The absolute bioavailability is relatively low, at 0.38% higher. The SNAC formulations consistently demonstrated blood concentrations higher than the EC_50_, underscoring their effectiveness in achieving the desired therapeutic effect [14]. The enhanced bioavailability observed with SNAC-PC salt can be attributed to the presence of lipophilic long chains within the PC that increase the hydrophobicity of the formulation and enhance its ability to penetrate the intestinal walls, leading to higher absorption into the bloodstream. This finding contributes to the salt-conversion strategy, which is evident as a beneficial tool for enhancing substance properties.

The PC molecule contains both a hydrophilic region and hydrophobic chains, essentially making it a surfactant. SNAC, itself, also possesses surfactant properties. Consequently, formulating diverse surfactants together can contribute to an additive effect on their activity, potentially leading to enhanced absorption [16]. These findings may demonstrate that SNAC salts, as well as their regeneration products in the gastrointestinal environment, do not impair the absorption of semaglutide. This offers a new potential strategy for enhancing the oral bioavailability by modifying the non-therapeutic agent, SNAC.

Studies investigating the oral absorption of drugs affected by SNAC have demonstrated improved penetration compared to other formulations. For example, a clinical study of vitamin B12 showed a significant increase in absolute absorption compared to the commercial product, from 2% to 5% [9]. Preclinical studies reached similar conclusions, although with slightly different values [17]. It is important to remember that the study model, as well as the specific drug being investigated, plays a role in the results, as the drug’s properties and size also affect its absorption. Semaglutide, being a long peptide, faces challenges in oral bioavailability due to its size, as well as stability and enzymatic degradation issues. Additionally, the mechanism of penetration enhancement by SNAC is not fully understood and may involve a unique pathway, as suggested in studies on semaglutide [13,18]. Consequently, following the presented proof of concept, it will be possible to further study the effect of the developed SNAC salts on bioavailability in additional models and clinical settings, as well as with other drugs.

In the current study, another important factor is that the formulations were prepared using weight ratios, resulting in variable amounts of SNAC in SNAC, SNAC-CH, and SNAC-PC. It should be noted that SNAC-CH contains 2/3 of the quantity of SNAC present in the original formulation, while SNAC-PC contains only 1/3 (see Section 2). Despite the lower amount of SNAC in SNAC-CH and SNAC-PC formulations, the formulations’ respective PK values not only reached, but even exceeded, those of the SNAC formulation. SNAC salt regenerated to SNAC in physiological acidic conditions, thus indicating that the conversion of SNAC to SNAC-CH or SNAC-PC can enable a lesser amount of the required SNAC. Formulating semaglutide with SNAC-CH or SNAC-PC, in a ratio that the SNAC weight equals that of the original formulation, may potentially, in optimal conditions, lead to even greater absorption and further enhancement of therapeutic benefits.

SNAC percentages in commercial products are more than half of the formulation. Therefore, the lowering of its percentage contributes to its accessibility [9,19], despite its being regarded as safe [20].

The professional literature has contained studies concerning the mechanism of SNAC to enhance permeability. It is not fully definitive, but suggested, that it is beneficial to reduce transepithelial electrical resistance (TEER) and increase membrane fluidity [21]. SNAC increases lipophilicity of the formulation and causes a local increase in pH via a buffering action [1,20]. A pre-clinical study in dogs using semaglutide formulated with SNAC shows that tablet location yields an impact on semaglutide penetration, with a 10-fold higher semaglutide concentration detected underneath the tablet, as compared to 6 cm at a further distance. This finding corroborates the suggested mechanism of semaglutide–SNAC tablet sedimentation in the stomach, slowly eroding and facilitating absorption [13]. In this study, the SNACs–semaglutide formulations were administered to rats after a period of fasting to eliminate the influence of the food digestion rate and the abundance of gastrointestinal factors on bioavailability.

## 4. Conclusions

While many strategies are known for broadening molecular libraries, the conversion of phenols into phenolate salts has been relatively underexplored. In this study, alkali salts (Na^+^), as well as ammonium salts (choline and PC), of SNAC were developed under optimal conditions, utilizing acid–base reactions. The SNAC salts were characterized and compared to the original SNAC using IR, NMR, and CHN analysis, confirming the successful formation of the SNAC salts. SNAC is a known and clinically used synthetic absorption enhancer that is considered safe. To exemplify the potential of using the developed phenolate salts as alternative absorption enhancers, a pharmacokinetic in vivo study was performed. Determining semaglutide plasma concentrations over time after oral administration in vivo revealed no detract on the PK of semaglutide using the synthesized SNAC salts and the potential of substituting SNAC with SNAC-PC to achieve better semaglutide absorption, while reducing the required amount of SNAC. This highlights the possibility of employing salt formation as a strategy to improve the delivery of proteins and peptides without affecting the therapeutic agent’s efficacy.

## Data Availability

The original contributions presented in the study are included in the article; further inquiries can be directed to the corresponding author.

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
