# Peer review of "The Synthesis of SNAC Phenolate Salts and the Effect on Oral Bioavailability of Semaglutide"

_molecules, 2024, doi:10.3390/molecules29163909_

Round 1

Reviewer 1 Report

Comments and Suggestions for Authors

 Professor Domb investigated the synthesis of SNAC phenolate salts, specifically SNAC-Choline (SNAC-CH), SNAC-Sodium (SNAC-Na), and SNAC-Phosphatidylcholine (SNAC-PC), and explored their potential application in improving the oral absorption of semaglutide. This is a simple and straightforward approach for SNAC derivatives and it is interesting for reader of  Molecules. It can be published after minor revision.

Comments:

Table 1. C, H, N, and O analysis data for SNAC-salts. SNAC, Choline salt of SNAC and phosphatidylcholine salt of SNAC need further purified. The C value of Experimental is high than Theory too much. The correct value between Experimental and Theory is less than 0.5%. 

Author Response

The authors are thankful to the reviewers for their useful comments

Reviewer 1:

Comment: Table 1. C, H, N, and O analysis data for SNAC-salts. SNAC, Choline salt of SNAC and phosphatidylcholine salt of SNAC need further purified. The C value of Experimental is high than Theory too much. The correct value between Experimental and Theory is less than 0.5%.

Response 1: Thank you for your insightful comment. We acknowledge that further purification of the choline salt from access choline, however, attempts to isolate the choline resulted in desalting of the SNAC and since choline is a safe component it was left in the final product. We have added an explanation within the manuscript for the C value between theoretical and experimental in the PC-SNAC: “The elemental analysis of the phospholipid salt, indicate excess phospholipid which is probably related to the large phospholipid residue that forms partial salts of the SNAC carboxylate and phenolate sites for salt formation and the difficulty to isolate the excess phospholipids from the salt product.”

Reviewer 2 Report

Comments and Suggestions for Authors

In the manuscript, the performance of N - [8- (2-hydroxybenzoyl) amino] octanoate sodium (SNAC) was attempted to be expanded through a strategy of converting it into phenolic salt structures such as SNA-CH, SNAC sodium (SNAC-Na), and SNAC phosphatidylcholine (SNAC-PC). These SNAC series phenolic salts were used for in vivo testing to evaluate the oral delivery of semaglutide. In in vivo experiments, drug absorption performance and their oral administration patterns were evaluated by measuring pharmacokinetic (PK) values and oral pharmacokinetics (PK) of SNAC phenols, and the results showed no damage to the initial SNAC PK values, which is expected to become a potential novel effective enhancement of oral absorption. However, at present, the manuscript needs to add some necessary experimental data, including comparing with other administration modes, evaluating whether oral administration under this compound formulation condition is the optimal administration mode, and further suggesting that the author does not provide detailed characterization of the compound drugs prepared by the author. It is recommended that the author characterize the structure of the series of SNAC phenolic salts as components of the compound formulation to confirm the author's animal experimental results. 

Author Response

The authors are thankful to the reviewers for their useful comments

Reviewer 2:

Comment: However, at present, the manuscript needs to add some necessary experimental data, including comparing with other administration modes, evaluating whether oral administration under this compound formulation condition is the optimal administration mode, and further suggesting that the author does not provide detailed characterization of the compound drugs prepared by the author. It is recommended that the author characterize the structure of the series of SNAC phenolic salts as components of the compound formulation to confirm the author's animal experimental results. 

Response 2: as stated in the title of this manuscript “The synthesis of SNAC phenolate salts and the effect on oral bioavailability of semaglutide“ the focus of this project was to assess the effect of SNAC salts on the oral bioavailability of this active compound. This is of clinical importance given the poor bioavailability of the SNAC-semaglutide that is currently marketed. The only comparison made is with an intravenous bolus of semaglutide administration which is required for the assessment of the “absolute bioavailability” value and other pharmacokinetic observations. Therefore, we find that the Reviewer's comment advising seeking potentially more “optimal administration modes” is beyond the scope of this manuscript.

Reviewer 3 Report

Comments and Suggestions for Authors

This study aimed to enhance the properties of SNAC by converting it into phenolate salts for potential use in improving oral absorption of semaglutide. The synthesis and successful confirmation of SNAC-CH, SNAC-Na, and SNAC-PC were achieved using analytical techniques. In vivo testing showed promising results for using these phenolate salts as effective enhancers for oral absorption without compromising therapeutic efficacy, offering a potential alternative to injectable peptides through oral delivery systems. Overall, this work is comprehensive and can be published if the authors could address the following issues:

Please provide a detailed description of the experiment for the elemental analysis of these compounds.

There is a typo right under 2.5. LC-MS analytical method. Also, please justify the gradient elution program design.

What were the MRM transitions optimized based on? Are there other conditions used?

For Figure 2 FTIR spectra, a table including the peak and the vibration mode could be more helpful in identifying these differences. 

Author Response

The authors are thankful to the reviewers for their useful comments 

Reviewer 3:

Comment: 1. please provide a detailed description of the experiment for the elemental analysis of these compounds.

  1. There is a typo right under 2.5. LC-MS analytical method. Also, please justify the gradient elution program design.
  2. What were the MRM transitions optimized based on? Are there other conditions used?
  3. For Figure 2 FTIR spectra, a table including the peak and the vibration mode could be more helpful in identifying these differences.

Response 3:

  1. Thank you for your comment. We have added a description in the methods section, as followed: “ Determination of C, H, N and O was performed using the Thermo Flash 2000 CHN-O Elemental Analyzer.“
  2. Extensive development work was conducted for the chromatography detection of semaglutide without relying on a documented method for the peptide, which does not exist in the literature. In the elution program, the transition from 40:60 to 20:80 allows for the proper elution and separation of semaglutide from other plasma ingredients and peptides that could interfere with the analysis, by increasing the competition between the mobile and stationary phases (RT = 6 minutes), after which the system is returned to 40:60. In addition, we have rephrased the gradient elution program paragraph: “Gradient elution mobile phases consisted of 0.1% formic acid in water (Phase A) and 0.1% formic acid in methanol (Phase B). A flow rate of 0.2 mL/min was applied with a linear gradient from 60% to 80% B over 3.5 minutes and then hold at 80% B for additional 4.5 minutes. The column was then equilibrated for 6 minutes at starting conditions prior to the injection of the next sample.”
  3. Comprehensive work was undertaken to develop the MS method. The reported MRM transitions were selected to optimize the sensitivity and selectivity of the method. We preferred the high transition due to its selectivity (many other peptides in the plasma can be fragmented into amino acids with a mass around 100-200). We have added a descriptive paragraph in the LC-MS method section: “Optimal detection conditions were determined by constant infusion of a 200 ng/mL solution of Semaglutide in methanol:water (8:2) using the integrated syringe pump (5 μL/min). Transitions were selected and their settings were determined using Analyst 1.7.1 Software in compound optimization mode.”
  4. We have added a table, as suggested.

O-H stretching (cm-1)

C–H stretching (cm-1)

Amide C=O stretching (cm-1)

carboxylate carbonyl stretching (cm-1)

C-N stretching (cm-1)

C=C stretching (cm-1)

C-O stretching (cm-1)

C–C bending (cm-1)

SNAC

3353

2992-2850

1637

1560

1495

1444-1421

1340-1310-1238-1147

1098- 1038

SNAC-Na

--

2928–2858

1618

1568

1452

1424-1406

1337-1300-1261-1215-1145

1030

SNAC-Ch

3246

2926–2854

1613

1557

1459

1403-1332

1250-1199-1145

1088-1055

SNAC-PC

--

2982–2852

1639

1590

1493

1446-1340

1242-1171

1090-1060

Round 2

Reviewer 2 Report

Comments and Suggestions for Authors

 After designing and supplementing the corresponding research content based on the following two published papers, the author resubmits the manuscript:

  Pharmacokinetics of Oral Cyanocobalamin Formulated With Sodium N-[8-(2-hydroxybenzoyl)amino]caprylate (SNAC): An Open-Label, Randomized, Single-Dose, Parallel-Group Study in Healthy Male SubjectsClinical TherapeuticsVolume 33, Issue 7, July 2011, Pages 934-945

The synthesis of SNAC phenolate salts and the effect on oral bioavailability of semaglutide Molecules 2023, 28, x. https://doi.org/10.3390/xxxxx

Author Response

Reviewer 2:

Comment:  After designing and supplementing the corresponding research content based on the following two published papers, the author resubmits the manuscript. 

Response 2: Thank you for helping to make the manuscript clearer and more accurate. The main focus of this study is to demonstrate the synthesis of SNAC salts, followed by the evaluation of their effect on oral bioavailability. Based on the reference you suggested (which was already included in the manuscript) and two additional new references, we have added the following:

To the introduction:

“A pharmacokinetics clinical trial comparing oral vitamin B12 and oral formulation with SNAC revealed a significant improved bioavailability for the SNAC formulation, as well as reduced Tmax.”

To the in vivo method:

“Absolute bioavailability values and other pharmacokinetic observations was assessed by comparison to intravenous bolus of semaglutide”.

To the discussion:

“Studies investigating the oral absorption of drugs affected by SNAC have demonstrated improved penetration compared to other formulations. For example, a clinical study of vitamin B12 showed a significant increase in absolute absorption compared to the commercial product, from 2% to 5%[9]. Preclinical studies reached similar conclusions, although with slightly different values [17]. It is important to remember that the study model, as well as the specific drug being investigated, play a role in the results, as the drug’s properties and size also affect its absorption. Semaglutide, being a long peptide, faces challenges in oral bioavailability due to its size, as well as stability and enzymatic degradation issues. Additionally, the mechanism of penetration enhancement by SNAC is not fully understood and may involve a unique pathway, as suggested in studies on semaglutide [13,18]. Consequently, following the presented proof of concept, it will be possible to further study the effect of the developed SNAC salts on bioavailability in additional models and clinical settings, as well as with other drugs”.
